# Enhancement of Medium-Carbon Steel Corrosion and Wear Resistance by Plasma Electrolytic Nitriding and Polishing

**Andrey Apelfeld** [1,2], **Anatoly Borisov** [1,2,*], **Ilya Dyakov** [3], **Sergey Grigoriev** [1], **Boris Krit** [1,2], **Sergei Kusmanov** [3], **Sergey Silkin** [3], **Igor Suminov** [1] **and Ivan Tambovskiy** [1,3]

1   Department of High-Efficiency Machining Technologies, Moscow State University of Technology "STANKIN", 127994 Moscow, Russia; apelfeld@yandex.ru (A.A.); s.grigoriev@stankin.ru (S.G.); bkrit@mail.ru (B.K.); ist3@mail.ru (I.S.); ramstobiliti@gmail.com (I.T.)
2   Moscow Aviation Institute, National Research University, 109383 Moscow, Russia
3   Department of Chemistry, Kostroma State University, 156005 Kostroma, Russia; igdyakov@mail.ru (I.D.); sakusmanov@yandex.ru (S.K.); longamin@mail.ru (S.S.)
*   Correspondence: anatoly_borisov@mail.ru; Tel.: +7-4959155441

**Abstract:** The influence of technological parameters of plasma electrolytic nitriding and polishing on the wear resistance and corrosion resistance of medium-carbon steel is considered. The morphology and roughness of the surface, phase composition and microhardness of the modified layer have been investigated. Wear resistance was studied under dry friction conditions with bearing steel as counter-body. It was found that plasma electrolytic polishing removes the loose part of the oxide layer and provides a two-fold decrease in surface roughness compared with untreated steel, and 2.8 times compared with the nitrided one. Combined processing at optimal technological parameters leads to an increase in microhardness up to 1130 HV, an increase in wear resistance by 70 times, and a decrease in the corrosion current density by almost 3 times in comparison with untreated steel.

**Keywords:** plasma electrolytic nitriding; plasma electrolytic polishing; surface roughness; microhardness; wear and corrosion resistance; medium carbon steel

## 1. Introduction

Plasma nitriding is a widely used technological process in the surface engineering of metal machine parts. When nitriding in a glow discharge plasma, to block the development of roughness during cathodic sputtering of the surface, plasma nitriding with an active screen is used [1–3]. In discharges with a hollow cathode [4,5] and arc discharges [6,7], nitriding is carried out in vacuum chambers with special screens and surface erosion by ion sputtering is also blocked. For cleaning and etching the surface, along with ion beams, neutral fluxes of atoms and molecules are used [8,9]. These technologies in [4–10] are intermediate before the application of hard ceramic coatings to cutting tools.

Similar problems of nitriding while maintaining the smoothness of the surface must be solved in plasma electrolytic nitriding (PEN). Over the past decades, plasma electrolytic technologies show an obvious tendency to expand in usage that is reflecting not only in the number of publications, but also in the more systematic description of these methods. A detailed classification of plasma electrolytic processes takes into account the type of metals and alloys to be treated, the structure of the near-electrode zone and methods of contacting the workpiece with the electrolyte, that is, immersion the treating surface in a solution or exposure in a spray [11]. New technologies for chemical synthesis, water purification, polymer degradation and hydrogen production are noted in the review [12]. One can note the anodic exfoliation of graphite to obtain of graphene-like structures [13,14]. Most of the researches were devoted to the surface treatment of metals and alloys in order to increase their hardness, wear and corrosion resistance. For example, the plasma electrolytic oxidation, widely used in modern industry, provides the formation of ceramic

coatings on valve metals and alloys [15,16]. Specific conditions in a vapor-gaseous envelope (VGE) surrounding the workpiece make it possible to significantly accelerate the diffusion processes of saturation of steels and titanium alloys with nitrogen, carbon, boron and their combinations [17–19]. In the industries of a number of countries, the plasma electrolytic polishing (PEP) of various alloys is successfully used [20]. The formation of nano-sized coatings and the production of nano-sized materials are considered as promising new technologies [21].

Plasma electrolytic nitriding provides a comprehensive improvement in the most characteristics of medium-carbon steels. Favorable combinations of oxide layer, nitride zone and martensite sublayer lead to increasing of hardness, fatigue strength [17], wear resistance [22,23] and corrosion resistance [24] of various iron-based alloys. Sources of nitrogen in aqueous solutions can be ammonia, carbamide and ammonium nitrate. The latter is distinguished by sufficient nitrogen potential, slower depletion and low cost. Its disadvantage is high oxidizing ability, which leads to the formation of a loose oxide layer and an increase in the roughness of the treated surface even with the anodic polarity of the workpiece. The surface quality can be improved by the subsequent PEP.

The aim of this work is to study the influence of the technological parameters of PEP on the characteristics of medium-carbon steel after anodic PEN. The research includes; studying the effect of voltage, temperature, composition of electrolytes and their hydrodynamic scheme of flow around the sample being polishing on the surface morphology and roughness as well as on corrosion and wear properties of have treated steel.

## 2. Materials and Methods

Cylindrical samples of medium-carbon steel (0.45 wt.% C) 10 mm in diameter and 15 mm in height were polished with sandpaper to the surface roughness of $R_a = 1.0 \pm 0.1$ μm and cleaned with acetone in ultrasonic bath. The PEN was carried out at the sample temperature of 850 °C in an aqueous solution of ammonium nitrate (5%) and ammonium chloride (15%) using cylindrical electrolyzer under conditions of longitudinal electrolyte flow around the sample [19]. In the upper part of the chamber-cathode, the electrolyte was being overflowed over the edge into the pallet, from where it was then pumped through the heat exchanger at flow rate of 2.6 L/min. The temperature of electrolyte cooled with tap water was maintained equal to $(25 \pm 2)$ °C at the inlet to the electrolyzer. During processing, the sample was connected to the positive pole of the DC power supply, and the working chamber to the negative one. After applying a voltage of 180 V, the sample was slowly lowered into the electrolyzer at a speed of 1 mm/s to a depth equal to its height. In this case, a continuous vapor-gas envelope began to form around the anode sample. After complete immersion and heating of the sample, the voltage dropped to 135 V, which corresponded to a surface temperature of 850 °C. The samples temperature was measured with the accuracy of 2% in the range from 400 to 1000 °C by chromel-alumel thermocouple connected to MS-8221 multimeter. The thermocouple was installed in a hole on the axis inside the sample at the distance of 2 mm from the heated surface. Diffusion saturation was carried out for 5 min, after which the samples were cooled in the electrolyte (quenched) from the saturation temperature when voltage was turned off. Then samples were washed with water and dried.

The subsequent plasma electrolytic polishing of the samples was performed under conditions of natural convection electrolyte or with a longitudinal vertical flow around the lateral surface of the sample at the flow rate of 0.8 L/min. The PEP duration was 1 min except when material removal was investigated (Section 3.3). In this case, nitrided samples were polished in turn in the same electrolyte under the same conditions for 5 min. The information given in Section 3.3 corresponds to the total value of the total weight of dissolved iron, and the removed part of the material of the samples polished in the same electrolyte. The electrolyte was subjected to production for up to 60 min.

Aqueous solutions of ammonium sulfate (3%) or ammonium chloride (3%), as well as aqueous solutions containing ammonium chloride (3%) with the additions of glycerol

(5%) or oxalic acid (5%) were used as electrolytes for the PEP. The voltage was varied in the range of 275–325 V. The samples were connected to the positive output of a power supply. The electrolyte temperature was varied in the range of 70–90 °C. The temperature of the sample during PEP corresponded to the boiling point of the electrolyte.

The phase composition of the surface layers was investigated using the X-ray diffractometer EMPYREAN (Malvern Panalytical, Malvern, UK) with Co $K_\alpha$ radiation and scanning in the $\Theta/2\Theta$ mode with a step of 0.1° and the rate of 1.25°/min. The cross-sectional structure of the surface layers was investigated using scanning electron microscopy Quanta 3D 200i (FEI Company, Eindhoven, The Netherlands).

To study the surface morphology, a Micromed MET optical metallographic microscope (Observing devices, St. Petersburg, Russia) with digital imaging of images was used.

The microhardness of the cross sections of the treatment sample was measured using a Vickers microhardness tester Falcon 503 (INNOVATEST Europe BV, The Netherlands) under a 0.1-N load. Then, 7 indentations were performed on each sample, and 5 points average value except both maximum and minimum values were used for hardness.

The surface roughness was measured with the TR-200 profilometer (TIME Group Inc. Beijing, China). The change in the weight of samples was determined on the CitizonCY224C electronic analytical balance (ACZET company, India) with an accuracy of ±0.0001 g after washing the samples with distilled water and subsequently drying. The weight of dissolved in electrolytes iron ions was determined by photocolorimetric method.

The corrosion resistance of samples was estimated at potentiodynamic polarization in sodium chloride (3.5%) solution using a Biologic SP-150 potentiostat-galvanostat in a standard three-electrode cell at the scan rate of 1 mV/s. Before corrosion testing, samples were cleaned with acetone in the ultrasonic bath for 5 min, then washed with distilled water and dried until their weight stabilized. Further, the surface of samples was insulated by a dielectric mask with a circular aperture with of 0.125 mm$^2$ area located at the distance of 2 mm from the lower edge of the sample. Graphite was used as the auxiliary electrode. The saturated silver chloride electrode served as the reference one. The working electrode (sample) was kept in sodium chloride (3.5%) solution for 60 min before testing to steady the constant value of corrosion potential. The corrosion potential and the corrosion current density were determined by Tafel's extrapolation of polarization curves with the aid of EC-Lab program.

The wear resistance of samples was studied with the aid of laboratory installation under dry friction conditions with bearing steel as counter-body at the normal load of 10 N and sliding speed of 1 m/s at the friction path of 1000 m at room temperature. Cylindrical surface of the samples was subjected to wear. Weight loss during testing was measured using a CitizonCY224C electronic balance after preliminary cleaning of the samples in acetone and subsequent drying. The characteristic morphology of the friction tracks was analyzed using a Micromed MET optical microscope.

### 3. Results

*3.1. X-ray Diffraction Analysis and Cross-Section Microstructure*

The results of X-ray diffraction analysis of the samples after PEN show the formation of FeO and $Fe_3O_4$ phases as a result of high-temperature oxidation of steel in the VGE, iron nitrides $Fe_4N$ as a result of diffusion of nitrogen, martensite and retained austenite as a result of quenching (Figures 1–3). The figures also show the results of the analysis of nitrided samples after their PEP under various conditions. The intensity of iron oxides peaks after PEP decreases, while nitrides peaks remain the same, which indicates partial removal of the oxides.

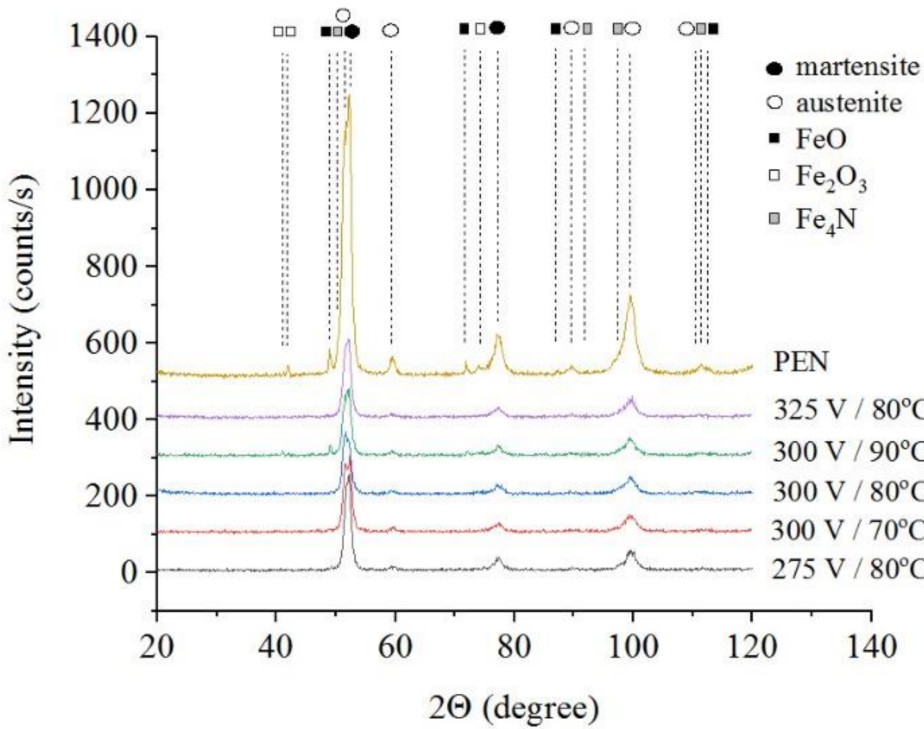

**Figure 1.** X-ray diffraction patterns of the steel surface layer after PEN in ammonium nitrate (5%) and ammonium chloride (15%) solution at 850 °C for 5 min and subsequent PEP in ammonium sulfate (3%) solution at different voltages and temperatures of electrolyte under natural convection.

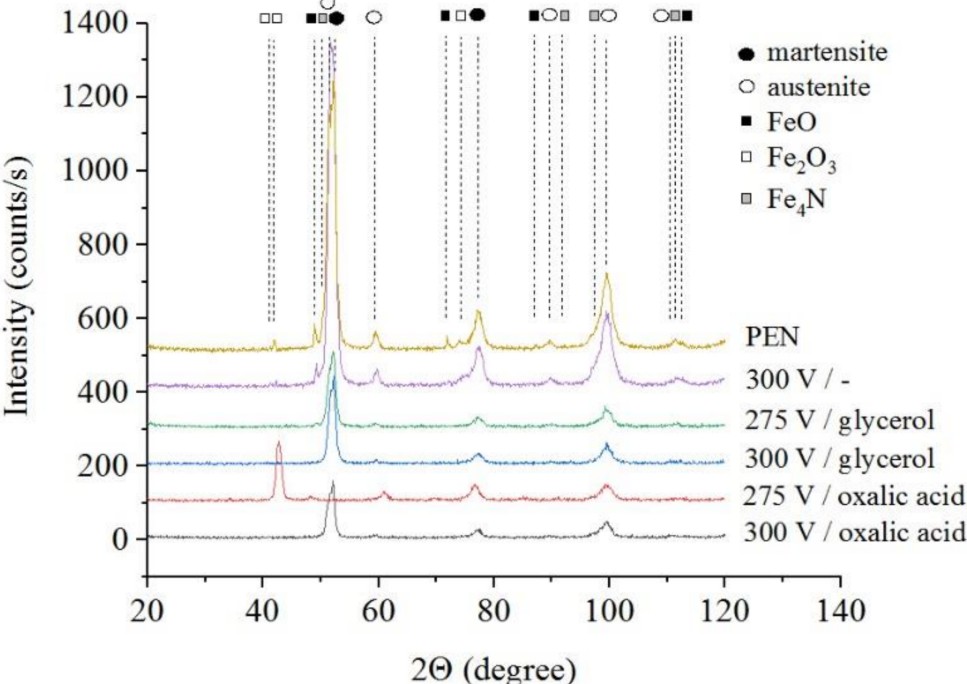

**Figure 2.** X-ray diffraction patterns of the steel surface layer after PEN and subsequent PEP in ammonium chloride (3%) solution with temperature of 80 °C without or with glycerol or oxalic acid additions (5% each) at different voltages under forced circulation of electrolyte.

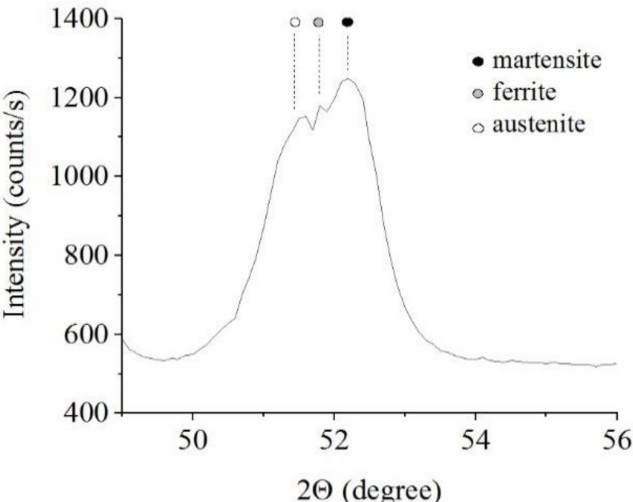

**Figure 3.** X-ray diffraction patterns of the steel surface layer after PEN.

According to SEM analysis, the structure of the treated surface is composed of the following layers (Figure 4):
− Oxide layer that contained FeO and $Fe_3O_4$;
− Nitride-martensite layer that contained $Fe_4N$, martensite and retained austenite;
− Martensite layer that contained martensite, retained austenite and solid solution of nitrogen);
− Initial pearlite-ferrite structure.

### 3.2. Surface Morphology

Morphology of the samples surface shows that PEN results in the removal of scratches, which remained after their polishing with sandpaper (Figure 5). The outer oxide layer formed after PEN contains delaminated areas. On the surface of nitrided steel, after PEP, the loose part of the oxide layer is removed, but this only occurs completely in some cases. The best polishing results are observed after PEP in ammonium chloride under natural electrolyte convection. When using a sulfate solution, the greatest removal of the loose layer occurs when using forced circulation of the electrolyte, as well as lowering its temperature. The quality of surface clarification during polishing is largely determined by the initial surface morphology. Even with significant material removal, a loose oxide layer can remain in the surface micro-cavities.

### 3.3. Material Removal and Surface Roughness

The weight loss of the samples (predominantly outer oxide layer) and the content of dissolved iron in the solution were measured. It was found that, all other things being equal, the flow of electrolyte around the samples leads to a more intensive removal of the oxide layer in comparison with the observed decrease in the sample mass under natural convection of electrolyte (Figure 6). In addition, more intense removal of material is observed during PEP in the ammonium chloride solution compared with the sulfate electrolyte. The concentration of iron (III) ions in solutions after the PEP is noticeably lower than the decrease in their weight (Figure 7). Therefore, the difference in these weights corresponds to the weight of oxygen contained in the removed oxide layer. In contrast to the weight loss, the amount of iron in the solution after the PEP without electrolyte circulation exceeds that when the electrolyte flows around the samples. This regularity is observed in both sulfate and chloride electrolytes.

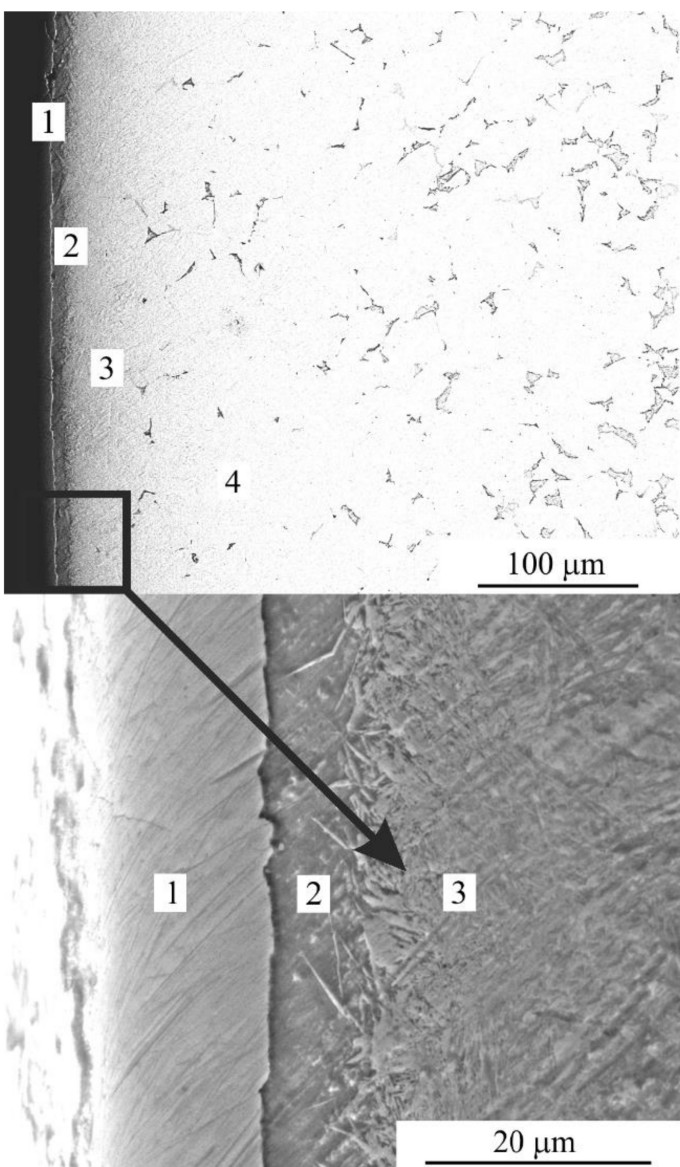

**Figure 4.** SEM image of cross-section of the steel surface after PEN. 1—oxide layer; 2—nitride-martensite layer; 3—martensite layer; 4—initial pearlite-ferrite structure.

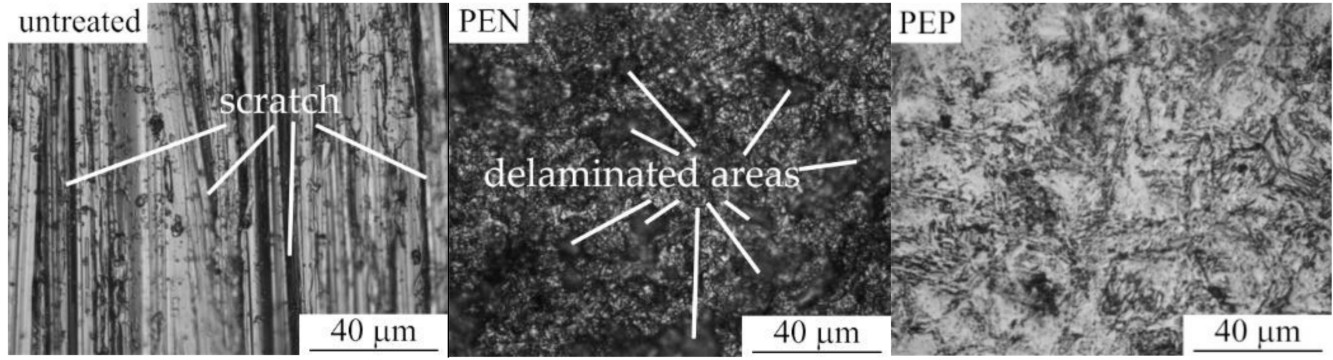

**Figure 5.** Surface morphology of untreated steel and after PEN as well as with subsequent PEP in ammonium chloride (3%) solutions under forced circulation of electrolyte at 300 V and 80 °C.

The intensity of iron dissolution during the PEP in the sulfate electrolyte correlates with the average current *I*. This tendency is observed when the electrolyte temperature changes from 70 °C to 90 °C at the constant voltage of 300 V both for the decrease in the sample weight (Figure 8) and for the intensity of anodic dissolution of iron (Figure 9). Additional heating of electrolyte increases VGE thickness and reduces current density. The effect of voltage at electrolyte temperature of 80 °C is less pronounced, but even in this case, the least dissolution is found at a voltage of 300 V, when the average current *I* is minimal.

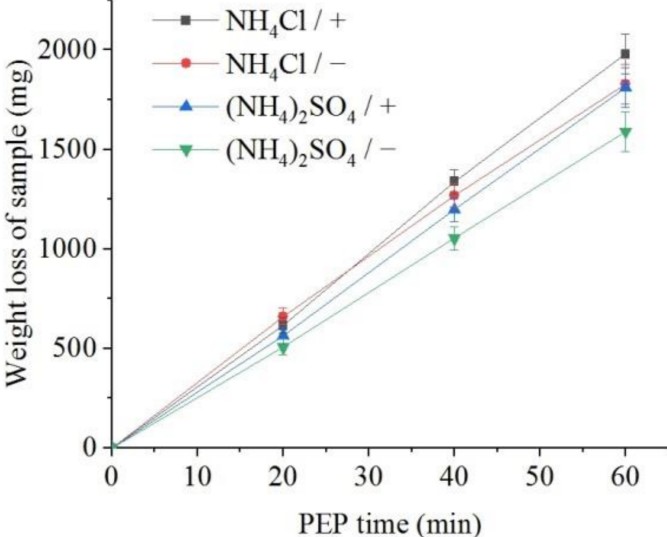

**Figure 6.** Dependence of the weight loss of the nitrided samples on the PEP duration at 300 V in ammonium sulfate (3%) or ammonium chloride (3%) solutions at their temperature of 80 °C under forced circulation (+) or natural convection (−) of electrolyte.

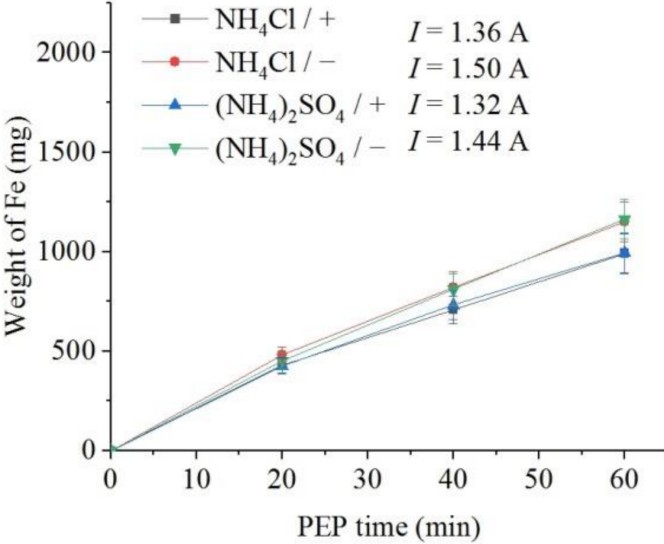

**Figure 7.** Dependence of iron weight dissolved from the nitrided samples on the PEP duration at 300 V in ammonium sulfate (3%) or ammonium chloride (3%) solutions at their temperature of 80 °C under forced circulation (+) or natural convection (−) of electrolyte.

The overall result of the PEP is a decrease in the weight of the samples and the roughness of their surface in all cases (Table 1). At the constant electrolyte temperature of 80 °C, the decrease of voltage to 275 V reduces surface roughness. Some correlation of change in roughness is observed between the decrease of electrolyte temperature and

removal of material. The best performance in terms of reducing roughness with the least material removal is obtained at a voltage of 300 V and a sulfate electrolyte temperature of 80 °C. The forced circulation of electrolyte leads to increase in the removal of material while maintaining the roughness, which excludes the feasibility of its use. In the case of the chloride electrolyte, there is a greater removal of material due to higher current density compared with the sulfate electrolyte, but significant decrease in roughness is achieved only with natural convection of the electrolyte.

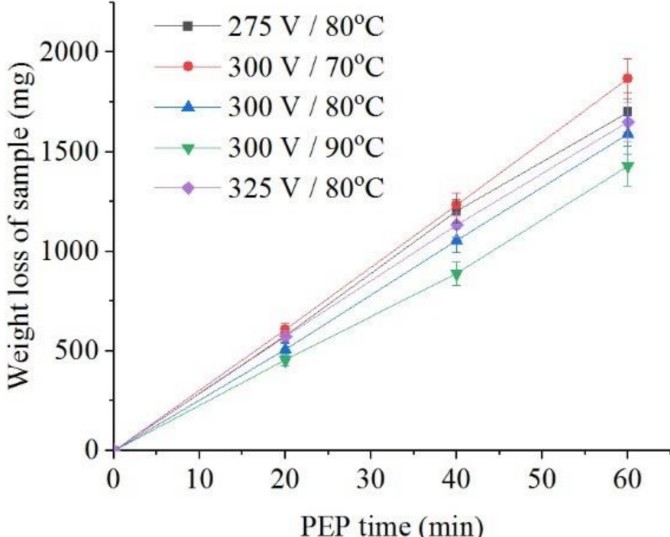

**Figure 8.** Dependence of the weight loss of the nitrided samples on the PEP duration in ammonium sulfate (3%) solution under natural convection of electrolyte conditions at different voltages and electrolyte temperatures.

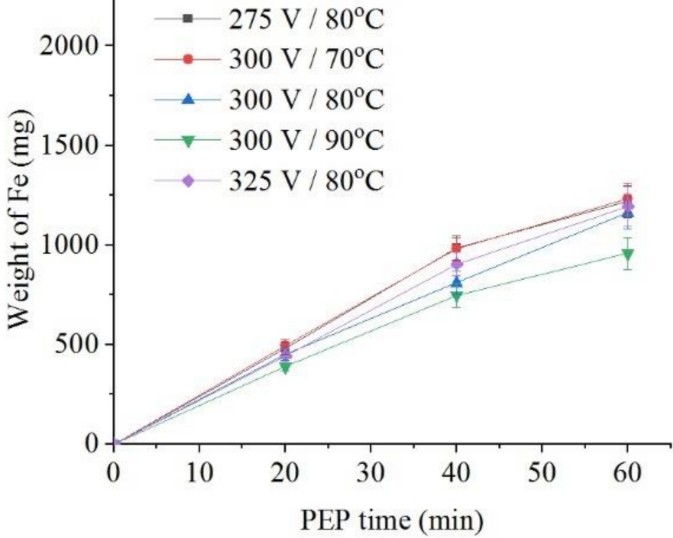

**Figure 9.** Dependence of iron weight dissolved from the nitrided samples on the PEP duration in ammonium sulfate (3%) solution under natural convection of electrolyte conditions at different voltages and electrolyte temperatures.

**Table 1.** Average current *I*, weight loss of samples $\Delta m$, surface roughness *R*a, corrosion current density *j*, friction coefficient $\mu$ and weight wear $\Delta m_{fr}$ before and after PEN with subsequent PEP in various electrolytes, at different voltages *U* and electrolyte temperatures $T_{el}$. The signs indicate the applied hydrodynamic scheme (HS): forced circulation (+) or natural convection ($-$) of electrolyte. The PEP processing time is 1 min.

| Electrolyte for PEP | *U*, V | $T_{el}$, °C | HS | *I*, A | $\Delta m$, mg | *R*a, μm | *j*, μA/cm² | $\mu$ | $\Delta m_{fr}$, mg |
|---|---|---|---|---|---|---|---|---|---|
| Before treatment | | | | | | $1.0 \pm 0.1$ | 23 | 0.61 | 13.8 |
| After PEN | | | | | | $1.4 \pm 0.2$ | 19 | 0.65 | 0.25 |
| Ammonium sulfate (3%) | 275 | 80 | $-$ | 1.7 | 28.7 | $0.5 \pm 0.1$ | 17 | 0.53 | 0.2 |
| | 300 | 70 | $-$ | 1.9 | 31.6 | $0.5 \pm 0.1$ | 13 | 0.54 | 0.2 |
| | 300 | 80 | + | 1.3 | 30.4 | $0.5 \pm 0.1$ | 14 | 0.61 | 0.4 |
| | 300 | 80 | $-$ | 1.4 | 27.0 | $0.5 \pm 0.1$ | 12 | 0.56 | 0.6 |
| | 300 | 90 | $-$ | 1.2 | 24.0 | $0.65 \pm 0.04$ | 10 | 0.54 | 0.5 |
| | 325 | 80 | $-$ | 1.3 | 27.7 | $0.6 \pm 0.1$ | 19 | 0.55 | 0.2 |
| Ammonium chloride (3%) | 300 | 80 | + | 1.4 | 33.2 | $0.7 \pm 0.1$ | 30 | 0.58 | 0.5 |
| | 300 | 80 | $-$ | 1.5 | 30.8 | $0.5 \pm 0.1$ | 16 | 0.54 | 0.2 |
| Ammonium chloride (3%) + glycerol (5%) | 275 | 80 | + | 1.5 | 23.5 | $0.7 \pm 0.1$ | 8 | 0.57 | 0.3 |
| | 300 | 80 | + | 1.4 | 22.8 | $0.7 \pm 0.1$ | 14 | 0.54 | 0.4 |
| Ammonium chloride (3%) + oxalic acid (5%) | 275 | 80 | + | 1.5 | 23.6 | $0.8 \pm 0.1$ | 26 | 0.54 | 0.4 |
| | 300 | 80 | + | 1.4 | 26.7 | $0.56 \pm 0.04$ | 23 | 0.52 | 0.3 |

The use of glycerol and oxalic acid leads to in the decrease of samples weight loss during anodic dissolution (Figures 10 and 11). Organic additives to ammonium chloride solution reduce the surface roughness only for glycerol at a voltage of 275 V and for oxalic acid at a voltage of 300 V.

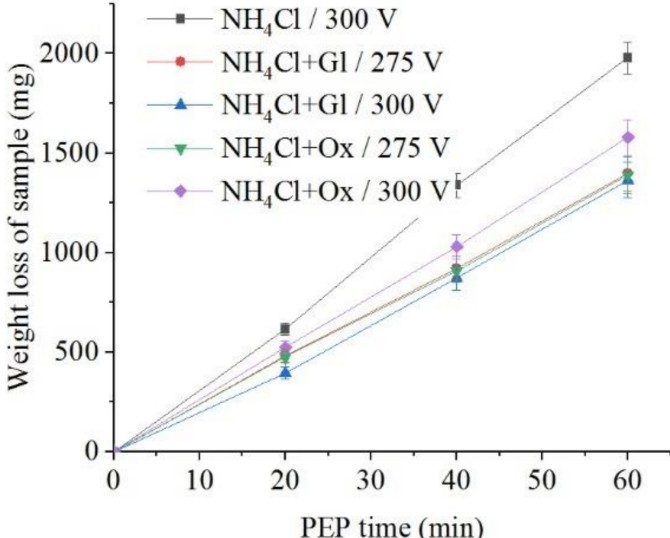

**Figure 10.** Dependence of the weight loss of the nitrided samples on the PEP duration in ammonium chloride (3%) solution at a temperature of 80 °C without or with glycerol or oxalic acid additions (5% each) at different voltages under forced electrolyte circulation.

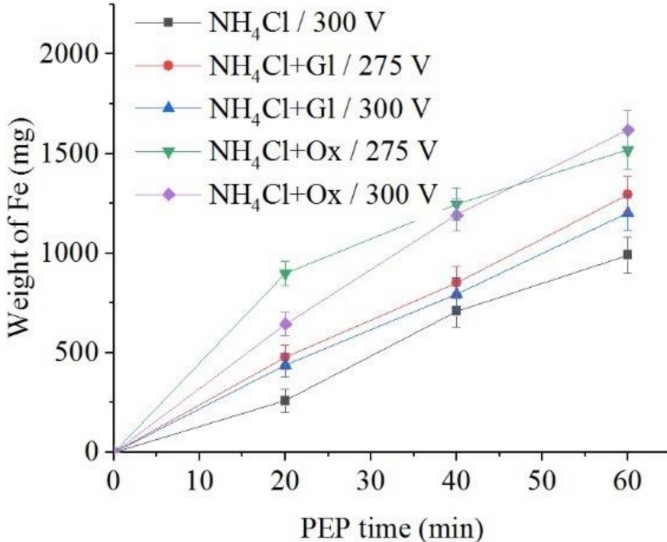

**Figure 11.** Dependence of iron weight dissolved from the nitrided samples on the PEP duration in ammonium chloride (3%) solution at a temperature of 80 °C without or with glycerol or oxalic acid additions (5% each) at different voltages under forced electrolyte circulation.

### 3.4. Microhardness

The microhardness of the surface layer of nitrided steel after PEP in the sulfate electrolyte without forced circulation practically does not change (within the measurement error), which confirms the invariability of the structure of the hardened layer (Figure 12). Similar results are observed for sulfate electrolyte at using different hydrodynamic schemes. The sample microhardness increases by about 100 HV when the electrolyte flows around the sample. Microhardness reduces by about 60 HV under natural convection (Figure 13). More noticeable decrease in microhardness (about 150 HV) of the outer part of diffusion layer is observed after PEP in the chloride solution under natural convection. The glycerol or oxalic acid addition to the ammonium chloride solution does not alter the microhardness of the surface layer (Figure 14).

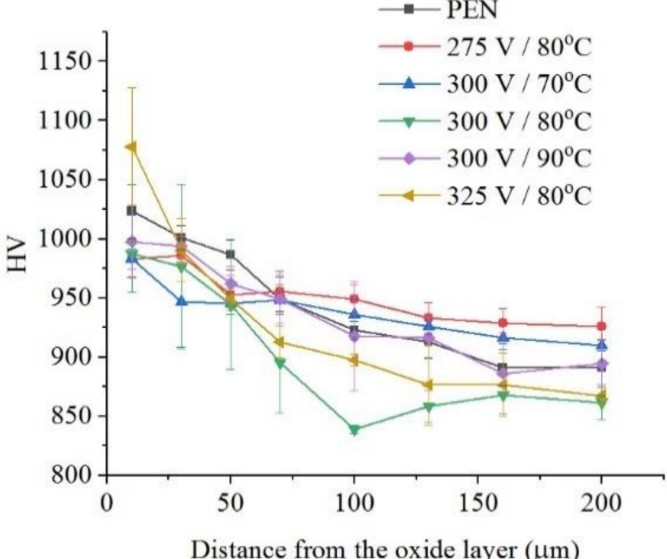

**Figure 12.** Distribution of microhardness in the surface layer of steel after PEN and subsequent PEP in ammonium sulfate (3%) solution under natural convection of electrolyte at different voltages and electrolyte temperatures.

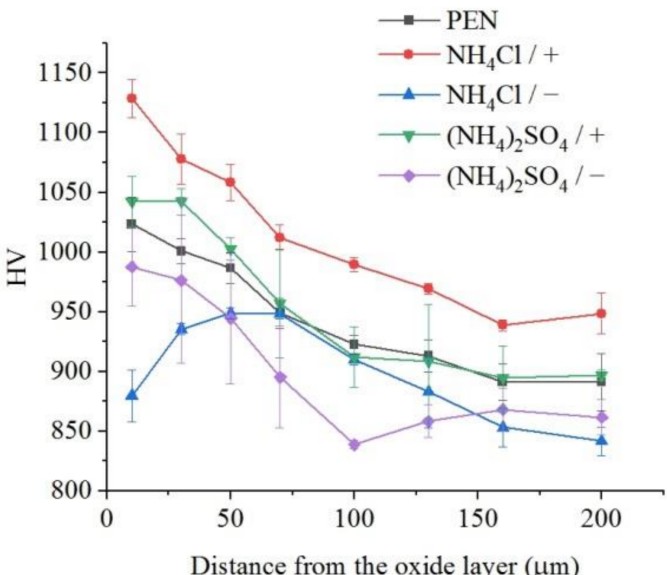

**Figure 13.** Distribution of microhardness in the surface layer of steel after PEN and subsequent PEP at a voltage of 300 V in ammonium sulfate (3%) or ammonium chloride (3%) solutions at a temperature of 80 °C under forced circulation (+) or natural convection (−) of electrolyte.

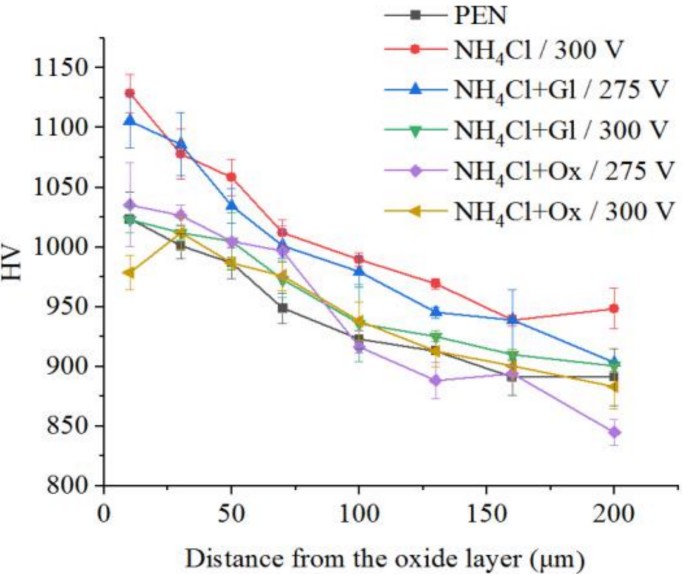

**Figure 14.** Distribution of microhardness in the surface layer of steel after PEN and subsequent PEP in ammonium chloride (3%) solution at a temperature of 80 °C without or with glycerol or oxalic acid additions (5% each) at different voltages under forced electrolyte circulation.

### 3.5. Corrosion Resistance

Corrosion tests show that the corrosion current density after PEN decreases from 23 to 19 µA/cm² (Table 1). The PEP of nitrided steel in sulfate electrolyte further improves corrosion resistance. The greatest decrease in the current density (by a factor of two) is observed after treatment at a voltage of 300 V and an electrolyte temperature of 90 °C with minimal material removal. Positive results in increasing the corrosion resistance after PEP in the chloride electrolyte are observed only under natural convection. Organic additives to the chloride electrolyte reduce the corrosion current density. However, the corrosion resistance in comparison with nitrided steel can be enhanced only in solutions containing glycerol.

*3.6. Wear Resistance*

It was found that PEN increases friction coefficient in comparison with untreated steel (Table 1). PEP provides a decrease in the coefficient of friction under all hydrodynamic conditions and electrolytes composition, making it less than that of untreated steel. The PEN permits to enhance the wear resistance of steel by a factor of 55. The additional PEP slightly improves or slightly worsens the wear resistance.

The analysis of the wear tracks shows that untreated sample is characterized by oxidative wear with presence of the areas of adhesive contact with counter-body under the given friction conditions (Figure 15a). After PEN the traces of the counter-body are not found on the friction surface, but traces of oxidative wear remain (Figure 15b). PEP increases the share of light areas without traces of oxidative wear (Figure 15c).

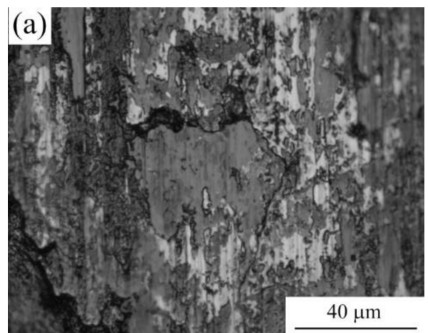 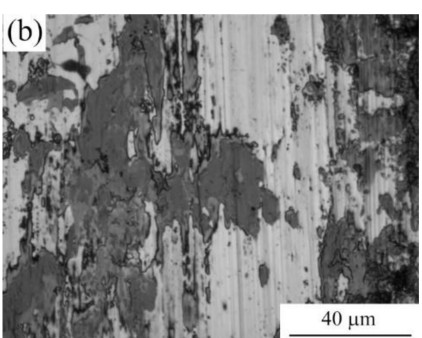 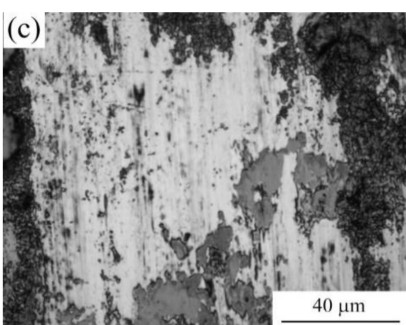

**Figure 15.** Morphology of the friction tracks on steel surface before treatment (**a**), after PEN (**b**) and subsequent PEP (**c**) in ammonium sulfate (3%) solution at a temperature of 80 °C at a voltage of 300 V under conditions of natural convection of electrolyte.

## 4. Discussion

The polishing the surface of the steel after its diffusion saturation is found to lead to the removal of the loose part of the outer oxide layer while maintaining the diffusion layer. The polishing rate is depended on the composition of the electrolyte. The electrolyte based on ammonium chloride promotes more intensive polishing than sulfate solution which is additionally involved in the passivation of the surface. Additionally, in the case of the chloride electrolyte, there is a greater removal of material due to higher current density compared with the sulfate electrolyte. Forced circulation of the electrolyte promotes a more uniform lightening of the surface, which is apparently associated with the equalization of the VGE thickness and the current density along the vertical of the sample by the electrolyte flow. The correlation between the intensity of dissolution and the current density is traced during heating of the electrolyte. Heating of electrolyte increases VGE thickness and reduces current density. The obtained dependences of the weight loss of the samples on the PEP duration are close to linear, which indicates the relative invariability of the structure of the layer being dissolved. In contrast to the weight loss, the amount of iron in the solution after the PEP without electrolyte circulation exceeds that when the electrolyte flows around the samples. This regularity is observed in both sulfate and chloride electrolytes and is explained by the correlation with the current density, which confirms the predominantly electrochemical mechanism of the anodic PEP [25]. Thus, natural convection of electrolyte favors the electrochemical dissolution of the material, while forced circulation of the electrolyte promotes greater destruction of the outer oxide layer. The latter fact does not fit into the framework of Faraday process of material removal.

The use of glycerol and oxalic acid leads to surface passivation by the products of the interaction of functional groups of organic molecules with the iron:

$$(n + 1)Fe^{3+} + mH_2O_{ads} + xOH^- \ (OH)_{ads} \rightarrow [Fe(Fe_nO_m) \cdot (OH)_x]_{ads} + 3(m + 1)H^+ \quad (1)$$

$$[Fe(Fe_nO_m) \cdot (OH)_x]_{ads} \rightarrow Fe_nO_m + Fe(OH)_x \quad (2)$$

The surface passivation is confirmed by X-ray diffraction analysis (Figure 2): the $Fe_2O_3$ peaks intensity increases significantly after the PEP in the solution of ammonium chloride with oxalic acid, while FeO, that being formed only at high-temperature oxidation, is practically absent. The PEP results in removal of oxide layer formed at PEN and forming of new oxide layer of composition $Fe_2O_3$. These processes are appeared in the increase of surface roughness at PEP in the solution of ammonium chloride with oxalic acid at a voltage of 275 V (Table 1). Organic additives to ammonium chloride solution reduce the surface roughness only for glycerol at a voltage of 275 V and for oxalic acid at a voltage of 300 V. In this case, the large decrease in samples weight occurs due to the inhibition of passivating effect of organic additives on the material surface.

The change in the microhardness of the surface layer can be explained by taking into account the processes of dissolution and oxidation of the material during PEP. A possible explanation for the decrease in microhardness when using a chloride electrolyte is the intensification of the electrochemical dissolution of the diffusion layer during natural convection of the electrolyte (Figure 7). This significant effect does not occur in the sulfate electrolyte due to the additional oxidation of the surface with sulfate ions. In the case of forced circulation of the electrolyte, the increase in microhardness is explained by the uniform distribution of the current density over the surface and the exclusion of its uneven etching. It can be assumed that the more intense electrochemical dissolution observed at natural convection of the electrolyte in contrast of circulation (Figure 7) leads to removing of the part of nitride-martensite layer along with the oxide layer. On the contrary, only small outer part of the diffusion layer including oxides is removed in conditions of flowing electrolyte, while the solid nitride-martensite layer is completely preserved, which is confirmed by X-ray diffraction analysis.

The decrease in the corrosion current density after PEN is due to the fact that the contact of the nitrided surface with the corrosive medium occurs through a modified layer saturated with iron oxides and nitrides, which play a positive role in the increase of corrosion resistance. The greatest decrease in the current density is observed at minimal material removal during PEP. The PEP of nitrided steel in sulfate electrolyte further improves corrosion resistance by passivating the surface. Organic additives to the chloride electrolyte reduce the corrosion current density in comparison with the use of pure ammonium chloride solution also due to additional surface passivation.

The change in surface morphology during PEP is reflected in the change in the coefficient of friction. The smoothing of microroughnesses promotes the sliding of the counterbody. The coefficient of friction correlates with surface roughness, which increased after PEN and decreased after PEP below the roughness of untreated steel.

## 5. Conclusions

1. Plasma electrolytic nitriding of medium-carbon steel increases surface roughness to 1.4 µm from 1.0 µm for untreated steel and provides formation of the nitride-martensite sublayer with the microhardness of 1050 HV. Plasma electrolytic polishing removes the loose part of the oxide layer formed after the PEN which leads to decreasing of the surface roughness. The nitride-martensite sublayer remains after 1 min PEP in ammonium sulfate aqueous solution (3%) or in ammonium chloride aqueous solution (3%). The recommended temperature for the polishing solution is 80 °C.

2. The weight loss of the samples during PEP occurs due to the predominant removal of the oxide layer. The transition of iron into electrolyte correlates with the current density, which varies with varying of voltage and electrolyte temperature that confirms the electrochemical nature of iron dissolution.

3. The highest microhardness of the medium-carbon steel surface layer is 1130 HV, and is reached after PEN in aqueous solution of ammonium nitrate (5%) and ammonium chloride (15%) at a sample temperature of 850 °C for 5 min, followed by quenching and PEP in ammonium chloride aqueous solution (3%) at a temperature of 80 °C for 1 min under longitudinal flow around the sample.

4. The corrosion current density of medium-carbon steel in sodium chloride solution (3.5%) could reduce from 23 $\mu A/cm^2$ for untreated steel to 19 $\mu A/cm^2$ after PEN and to 8 $\mu A/cm^2$ after PEP in ammonium chloride solution (3%) at 275 V for 1 min. Increasing of corrosion resistance is associated with the formation of iron oxides and nitrides during the PEN, as well as surface passivation during PEP under conditions of natural convection of electrolyte with minimal material removal.

5. The maximal decrease in the dry friction coefficient with bearing steel as counter-body (normal load of 10 N, sliding speed of 1 m/s) from 0.61 to 0.52 occurs after PEN with quenching and PEP in ammonium chloride solution (3%) at 300 V with addition of oxalic acid (5%). The PEP significantly reduces the share of areas of adhesive contact with the counter-body on the friction tracks.

6. Wear rate after combined treatment decreases by almost 70 times in comparison with untreated steel due to the formation of hard nitride-martensite layer after PEN and good running-in of the oxide layer dense part after PEP.

**Author Contributions:** Conceptualization, A.B. and S.K.; methodology, S.K., B.K. and I.S.; validation, S.G.; investigation, A.A., I.D., S.S. and I.T.; writing—original draft preparation, S.K.; writing—review and editing, A.B. and S.K.; project administration, S.G.; All authors have read and agreed to the published version of the manuscript.

**Funding:** This research was carried out with the financial support of the Russian Science Foundation within the framework of scientific project No. 21-79-30058. The study was carried out on the equipment of the Center of collective use of MSUT "STANKIN".

**Institutional Review Board Statement:** Not applicable.

**Informed Consent Statement:** Not applicable.

**Data Availability Statement:** Not applicable.

**Conflicts of Interest:** The authors declare no conflict of interest.

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
