# Peer review of "Enhancement of Medium-Carbon Steel Corrosion and Wear Resistance by Plasma Electrolytic Nitriding and Polishing"

_metals, doi:10.3390/met11101599_

Round 1

Reviewer 1 Report

 This paper examined the influence of technological parameters of plasma electrolytic nitriding and polishing on the wear resistance and corrosion resistance of medium-carbon steel. This paper is well organized and has useful scientific values.  

On the other hand, there are a few questions that reviewer would ask author to consider and modify this paper.  

(1) Figure 4 of cross-section of the steel surface after PEN shows the surface was covered by the oxide layer, which thickness was about 10 µm. In the contrast, Figure 1 of X-ray diffraction patterns in this sample shows mainly martensite and austenite peaks, Fe2O3 peaks are slight. Why Fe2O3 peaks are slight?

In addition, it is better that there are SEM images of cross-section of the steel surface after PEP.

(2) In Figure 13 and 14 of distribution of microhardness in the surface layer,it is seems that all samples will have the same hardness at the distance from the oxide layer 200μm. Because PEP can remove only the surface oxide layers, not effects on the inside of the medium-carbon steel.

Author Response

 This paper examined the influence of technological parameters of plasma electrolytic nitriding and polishing on the wear resistance and corrosion resistance of medium-carbon steel. This paper is well organized and has useful scientific values.

On the other hand, there are a few questions that reviewer would ask author to consider and modify this paper.

(1) Figure 4 of cross-section of the steel surface after PEN shows the surface was covered by the oxide layer, which thickness was about 10 µm. In the contrast, Figure 1 of X-ray diffraction patterns in this sample shows mainly martensite and austenite peaks, Fe2O3 peaks are slight. Why Fe2O3 peaks are slight?

The peaks of iron oxides are insignificant, since this layer should be considered more correctly as a layer of steel saturated with oxides. Chemically pure oxide layers are formed and peel off almost immediately during processing due to different thermal expansion, since they are not strong.

In addition, it is better that there are SEM images of cross-section of the steel surface after PEP.

The cross-section of the steel surface after PEP is no different from after PEN. Changes are visible only on the morphology of the surface itself.

(2) In Figure 13 and 14 of distribution of microhardness in the surface layer,it is seems that all samples will have the same hardness at the distance from the oxide layer 200μm. Because PEP can remove only the surface oxide layers, not effects on the inside of the medium-carbon steel.

Yes, PEP does not affect the hardened layer, but only the surface layer of oxides. In fig. 13 and 14 show the microhardness values of the hardened layer without oxide.

Reviewer 2 Report

I have a few comments and complementary suggestions to authors about the experimental procedure:

i) A hardness of ≈700-720 HV (equivalent to 60-61 HRC) is supposed to be presented for a martensite with 0.45 wt.% C. So, by comparing this hardness value (720 HV) with those of the obtained hardness profiles, supposedly nitriding depths higher than 200 mm were obtained after 850 oC/5 min, after quenching, is not it? So, authors should confirm if the treatment is diffusion-controlled, explaining theoretically with references, or estimating it through the treatment kinetics data.

ii) Authors have stated that “The PEN was carried out at the sample temperature of 850 °C in an aqueous solution of ammonium nitrate (5%) and ammonium chloride (15%) using cylindrical electrolyser under conditions of longitudinal electrolyte 73 flow around the sample [19].”. So, I think that details about the electrolytic solution used for PEN, as above indicated, mainly its boiling point should be presented. In addition, comments about how its vaporization was effectively avoided, if so, since 850 oC is considerably a very high temperature, thus an eventual vapor film around the studied surface would tend to acts as a thermal insulator during the cooling step along the performed quenching, supposedly bringing a non-homogeneous martensite distribution all over the treated surface.

iii) Authors also stated that “…After applying the voltage of 180 V from DC power supply, the samples were immersed in the electrolyte followed by voltage correction to achieve the specified sample temperature.”. So, I kindly ask to the authors to clearly confirm in the manuscript text the actual principle used to the sample heating. This information is missing, even it seems to be a fool question, at least for readers that know the bases of the performed treatment.

Author Response

I have a few comments and complementary suggestions to authors about the experimental procedure:

i) A hardness of ≈700-720 HV (equivalent to 60-61 HRC) is supposed to be presented for a martensite with 0.45 wt.% C. So, by comparing this hardness value (720 HV) with those of the obtained hardness profiles, supposedly nitriding depths higher than 200 mm were obtained after 850 oC/5 min, after quenching, is not it? So, authors should confirm if the treatment is diffusion-controlled, explaining theoretically with references, or estimating it through the treatment kinetics data.

The hardness of the surface layer was measured after nitriding followed by quenching. Higher values ​​of hardness than for martensite steel with 0.45% C (720) are associated with the presence of diffusion nitrogen in the surface layer. The distribution of hardness correlates with the kinetic distribution of the concentration of diffused nitrogen. In addition, a feature of this technology is the formation of compressive stresses on the surface, which also leads to the hardening of the surface layer.

ii) Authors have stated that “The PEN was carried out at the sample temperature of 850 °C in an aqueous solution of ammonium nitrate (5%) and ammonium chloride (15%) using cylindrical electrolyser under conditions of longitudinal electrolyte 73 flow around the sample [19].”. So, I think that details about the electrolytic solution used for PEN, as above indicated, mainly its boiling point should be presented. In addition, comments about how its vaporization was effectively avoided, if so, since 850 oC is considerably a very high temperature, thus an eventual vapor film around the studied surface would tend to acts as a thermal insulator during the cooling step along the performed quenching, supposedly bringing a non-homogeneous martensite distribution all over the treated surface.

A feature of this process is the formation during the anodic treatment of a continuous vapor-gas envelope around the sample-anode with a thickness of several hundred micrometers. This vapor-gas envelope separates the metal anode from the electrolyte, creating a very high resistance and leading to the heating of this shell to high temperatures (from 400 to 1100 °C). It is from this envelope that part of the heat goes to the sample-anode, heating it to the specified processing temperature. Another part of the heat goes into the electrolyte solution, causing it to evaporate from the boundary between the solution and the vapor-gas envelope. The boiling point of the electrolyte solution is slightly more than 100 °C, but due to the small area of its contact with the vapor-gas envelope and additional cooling, the volume of the evaporated electrolyte is not large (the evaporation rate is about 1 ml / min). Circulation of a cooled electrolyte to a sample with a longitudinal flow makes it possible to control the constant thickness of the vapor-gas envelope along the sample. During quenching, the shell collapses and the heated metal contacts the cooled electrolyte. The results of measuring the hardness along the sample showed a constant value within the measurement error.

iii) Authors also stated that “…After applying the voltage of 180 V from DC power supply, the samples were immersed in the electrolyte followed by voltage correction to achieve the specified sample temperature.”. So, I kindly ask to the authors to clearly confirm in the manuscript text the actual principle used to the sample heating. This information is missing, even it seems to be a fool question, at least for readers that know the bases of the performed treatment.

The manuscript has been changed:

“During processing, the sample was connected to the positive pole of the DC power supply, and the working chamber to the negative one. After applying a voltage of 180 V, the sample was slowly lowered into the electrolyser at a speed of 1 mm/s to a depth equal to its height. In this case, a continuous vapor-gas envelope began to form around the anode sample. After complete immersion and heating of the sample, the voltage dropped to 135 V, which corresponded to a surface temperature of 850 °C.”

Reviewer 3 Report

The manuscript “Enhancement of medium-carbon steel corrosion and wear resistance by plasma electrolytic nitriding and polishing” by A. Apelfeld et al. deals with a study of electrolytic treatment of steel containing 0.45% C in various experimental conditions in view of improving the wear resistance of the surface, resistance to corrosion or surface hardness.

The manuscript presents a wealth of data and contains impressive results, such as an increase by 70 times of the wear resistance of treated samples.

There are however problems that the readers of this manuscript will encounter:

-there are figures that do not succeed to prove the authors affirmations, as e.g. fig 5 or fig 15. It is impossible to  judge for example the quality of the surface from fig 5.

- the presentation of experimental results is convoluted, with too many experimental parameters varied in the same time (ammonium chloride or sulphate concentration, experimental temperature,  addition of other chemicals…). The reader could be baffled by this amount of data presented in a mixed manner. Some conditions favor corrosion resistance, some other increase wear resistance, some increase the microhardness….in the end, the reader can not understand the exact purpose of the research.

There are some questions that rose during the lecture of the manuscript:

  1. Why were the samples heated at 850 C? By checking the iron-carbon diagram, it seems that phase changes could be possible for this type of steel close to this temperature. Is this an envisaged possibility?
  2. Which is the concentration in Ni or Mn of the steel? Why do the authors speak of passivation? Why would an iron oxide layer be called passivation layer?
  3. There are some values in case of results presented in figs 6-14 that are very close. How many samples of the same type have been tested? Which is the error for these experiments? Are the results statistically relevant?

Author Response

    Why were the samples heated at 850 C? By checking the iron-carbon diagram, it seems that phase changes could be possible for this type of steel close to this temperature. Is this an envisaged possibility?

The treatment was carried out at a temperature of 850 ° C, since at this temperature there is an intense diffusion of nitrogen, and during quenching from this temperature, a martensitic transformation occurs.

    Which is the concentration in Ni or Mn of the steel? Why do the authors speak of passivation? Why would an iron oxide layer be called passivation layer?

The concentration of Mn was up to 0.8%, and Ni up to 0.45%.

"Passivation" is a term taken from electrochemistry. It is used to denote a decrease in the electrochemical activity of processes, in particular, with the formation of an oxide film on a metal surface. This also applies to this work.

    There are some values in case of results presented in figs 6-14 that are very close. How many samples of the same type have been tested? Which is the error for these experiments? Are the results statistically relevant?

Testing was carried out on 3 samples for one experimental point. Statistical processing was carried out and the data are statistically reliable.

The text was revised again to correct any inaccuracies in the English language.

Round 2

Reviewer 3 Report

I do not consider that the authors answered satisfactory to my comments.

There are figures that do not offer information on the differences between various experimental conditions: see Fig 5

The presentation of experimental results is not clear, as there are too many parameters varied. One solution would be to present them in batches: one type of immersion solution  vs experimental parameters

There are no error bars on numerous figures. Some points are very close and could be statistically irrelevant

Author Response

  1. There are figures that do not offer information on the differences between various experimental conditions: see Fig 5

Thanks for the relevant comment. Indeed, the 11 SEM images in Figure 5 are redundant. To demonstrate the main differences in surface morphology under different experimental conditions in the corrected version, we present only 3 most typical SEM images of the surface before treatment, after PEN, and after the subsequent PEP.

2.The presentation of experimental results is not clear, as there are too many parameters varied. One solution would be to present them in batches: one type of immersion solution  vs experimental parameters

Thanks for the valuable suggestion for presenting the results. Now graphical information is presented by groups of compared parameters with subsequent analysis in the discussion section. Probably present them in batches: one type of immersion solution vs experimental parameters will make the text clearer, but we need a lot of time for that.

3.There are no error bars on numerous figures. Some points are very close and could be statistically irrelevant

In the updated version, we have added error bars in Fig. 6-11. We agree that some points are very close to each other and may be statistically irrelevant. We took this into account when discussing and drawing conclusions.

All corrections in the new version of the article, including the English correction, are highlighted

Round 3

Reviewer 3 Report

The authors performed most modifications suggested by this  reviewer. In my opinion, the manuscript increased in scientific soundness by inserting the statistical analysis and and can be accepted for publication